# Racial and ethnic variation in COVID-19 care, treatment, and outcomes: A retrospective cohort study from the MiCOVID-19 registry

Nadia R. Sutton[1]*, Sheria G. Robinson-Lane[2], Raymond Y. Yeow[1], Heather A. Chubb[3], Tae Kim[4], Vineet Chopra[4]

1 The Division of Cardiovascular Medicine, Department of Medicine, Michigan Medicine, Ann Arbor, Michigan, United States of America, 2 Department of Systems, Populations and Leadership, University of Michigan School of Nursing, Ann Arbor, Michigan, United States of America, 3 Department of Orthopaedic Surgery, University of Michigan, Ann Arbor, Michigan, United States of America, 4 The Patient Safety Enhancement Program, Division of Hospital Medicine, Department of Medicine, Michigan Medicine, Ann Arbor, Michigan, United States of America

* nadiaraz@med.umich.edu

**Data Availability Statement:** The data use agreements among the hospitals participating in

## Abstract

### Background

Racial and ethnic disparities in COVID-19 outcomes exist, but whether in-hospital care explains this difference is not known. We sought to determine racial and ethnic differences in demographics, comorbidities, in-hospital treatments, and in-hospital outcomes of patients hospitalized with COVID-19.

### Methods and findings

This was a cohort study using MiCOVID-19, a multi-center, retrospective, collaborative quality improvement registry, which included data on patients hospitalized with COVID-19 across 38 hospitals in the State of Michigan. 2,639 adult patients with COVID-19 hospitalized at a site participating in the MiCOVID-19 Registry were randomly selected. Outcomes included in-hospital mortality, age at death, intensive care unit admission, and need for invasive mechanical ventilation by race and ethnicity. Baseline comorbidities differed by race and ethnicity. In addition, Black patients had higher lactate dehydrogenase, erythrocyte sedimentation rate, C-reactive protein, creatine phosphokinase, and ferritin levels. Black patients were less likely to receive dexamethasone and remdesivir compared with White patients (4.2% vs 14.3% and 2.2% vs. 11.8%, p < 0.001 for each). Black (18.7%) and White (19.6%) patients experienced greater mortality compared with Asian (13.0%) and Latino (5.9%) patients (p < 0.01). The mean age at death was significantly lower by 8 years for Black patients (69.4 ± 13.3 years) compared with White (77.9 ± 12.6), Asian (77.6 ± 6.6), and Latino patients (77.4 ± 15.5) (p < 0.001).

### Conclusions

COVID-19 mortality appears to be driven by both pre-hospitalization clinical and social factors and potentially in-hospital care. Policies aimed at population health and equitable

MiCOVID-19 prohibit sharing of the full registry data outside of the Coordinating Center, given hospital-specific performance and events are directly identifiable. Requests for the data set used to perform these analyses (without hospital-specific identifiers) may be submitted to Elizabeth McLaughlin (emcnair@umich.edu) to be routed to the Michigan Hospital Medicine Safety (HMS) Consortium Data, Design and Publications Committee for approval of distribution. The code used to generate the findings is available at https://github.com/hdchubb/MARCQI-MICOVID19-SDOH-Manuscript.git. This information has been added to the paper.

**Funding:** Blue Cross Blue Shield of Michigan (BCBSM) and Blue Care Network as part of the BCBSM Value Partnerships program. The funder, the Blue Cross Blue Shield of Michigan and Blue Care Network, provided support in the form of salaries for authors (H.A.C. and T.K.), but did not have any additional role in the study design, data collection and analysis, decision to publish, or preparation of the manuscript. The specific roles of these authors are articulated in the 'author contributions' section. The National Institute on Aging 1K76AG064426-01A1 (N.R.S.) and K01AG06542001A1 (S.G.R.).

**Competing interests:** N.R.S. has received honoraria for speaking from Zoll Medical and is a Cordis and Philips advisory board member. This does not alter our adherence to PLOS ONE policies on sharing data and materials. V.C. has received royalties from Wolters Kluwer publishing, Oxford University publishing, and Lippincott for sales of academic textbooks. All other authors have no disclosures.

application of evidence-based medical therapy are needed to alleviate the burden of COVID-19.

## Introduction

In late 2019, reports regarding severe acute respiratory syndrome coronavirus 2 (SARS-CoV-2), which causes COVID-19, emerged from Wuhan, China. Spread of SARS-CoV-2 has subsequently fueled a global pandemic that has impacted over 185 countries, with over 6.3 million deaths worldwide [1]. Clinical manifestations of COVID-19 vary widely and may include upper respiratory symptoms, fever, headache, and gastrointestinal symptoms [2–4]. During the early months of the pandemic in the United States, a multitude of studies reported racial and ethnic disparities in outcomes among US patients infected with SARS-CoV-2 [5–10]. Specifically, several studies reported that Black and Latino patients were more likely to acquire COVID-19, require hospitalization, intensive care unit (ICU)-level care, and experience mortality than White patients [11–17]. Subsequent studies continued to report a growing gap in infections and deaths between Black, Indigenous, Asian, Latino, and other people of color compared to White patients [18, 19]. Yet, reasons underpinning this divide are poorly understood.

Several hypotheses explicating racial and ethnic disparities observed in COVID-19 outcomes have been postulated, including attribution to consequences of pre-existing medical comorbidities, socioeconomic factors, inability to effectively quarantine, and access to healthcare resources [7, 20–27] Some studies have suggested that differences in outcomes may relate to access to care or variation in hospital care [28, 29]. Despite these theories, few studies have examined differences in hospital care processes as a potential moderator of adverse outcomes in patients with COVID-19. To improve patient care and clinical outcomes, hospitals across the State of Michigan joined to collect data on patients hospitalized with COVID-19, forming the MiCOVID-19 Consortium. Using this dataset, we sought to better understand whether and how clinical presentation, care processes, and clinical outcomes varied by race and ethnicity. Our aim was to identify opportunities for improving management and optimizing resource allocation to improve outcomes in patients with COVID-19.

## Methods

MiCOVID-19 is a multi-center collaborative quality improvement [30–32]. A total of 38 hospitals (out of a total of 92 non-critical access, non-federal hospitals in Michigan), volunteered to participate in the registry. Although data are prospectively collected, we performed a retrospective analysis of previously collected data for this analysis. Hospitals contributing data to the registry were members of other collaborative quality initiatives. The study was deemed "not regulated" by the University of Michigan Institutional Review Board (HUM 00179611). The data use agreements among the hospitals participating in MiCOVID-19 prohibit sharing of the full registry data outside of the Coordinating Center, given hospital-specific performance and events are directly identifiable. Requests for the data set used to perform these analyses (without hospital-specific identifiers) may be submitted to Elizabeth McLaughlin (emcnair@umich.edu) to be routed to the Michigan Hospital Medicine Safety (HMS) Consortium Data, Design and Publications Committee for approval of distribution.

Patients discharged from a participating site between February 19, 2020 and November 19, 2020 were used in this analysis. At each hospital, trained clinical abstractors (often registered

nurses) collected data from the medical records of a random sample of adult (>18 years of age) patients using structured templates. Abstractors were aware that data would be utilized for quality improvement but did not have knowledge of the specific study objectives. For hospitals unable to abstract data for all COVID-19 hospitalizations, a pseudo-randomization procedure was utilized to select patients for detailed abstraction. Patients were randomly selected for full data abstraction as follows: first, positive cases were sorted by day of admission (Monday through Sunday). Then, for each day, the lowest minute of discharge (00–59) was used to select patients for detailed abstraction. Information on presentation (e.g., comorbidities, medications), clinical findings (laboratory results), treatment modalities, in- and post-hospital outcomes were collected. Patients were included in the registry if they tested positive for SARS-CoV-2 (via reverse-transcriptase polymerase chain reaction). We excluded patients who were admitted as persons under investigation, but ultimately did not have a positive test for SARS-CoV-2. Patients who were pregnant, transitioned to hospice within 3 hours of hospital admission, or were discharged against medical advice were also excluded. All data were entered into a registry (MiCOVID-19) using a structured, prospectively defined data collection template.

Data on patient race and ethnicity was abstracted directly from medical records. The following categories were included: Non-Hispanic, Hispanic or Latino, American Indian or Alaskan Native, Asian, Black or African American, Native Hawaiian or Pacific Islander, White or Caucasian, other, or unknown (if not indicated in the medical record).

Since we were interested in understanding whether differences in presentation and treatment were present by race or ethnicity, we examined whether use of therapies such as inpatient angiotensin-converting enzyme inhibitors, antibiotics, antivirals, hydroxychloroquine, interleukin-6 inhibitors, corticosteroids, and vasopressors varied by race or ethnicity. Advance directives at the time of presentation refers to patient wishes for no cardiopulmonary resuscitation, no intubation or mechanical ventilation, no non-invasive positive pressure ventilation, or no central line.

## Statistical analysis

Descriptive statistics including mean, standard deviation (SD), and proportions were used to summarize the data. For comparisons where one of the groups had a sample size less than 30, median and interquartile range (IQR) with a Kruskal-Wallis p-value are reported. Characteristics between racial and ethnic groups were compared using one-way Analysis of Variance (ANOVA) for continuous variables and Pearson's Chi-Squared test for categorical variables. An overall alpha-level of 0.05 was used to determine statistical significance and all statistical tests are two-sided. All data was analyzed using SAS software version 9.4 (SAS Institute Inc., Cary, NC, USA). The full statistical code used to generate the findings is available at https://github.com/hdchubb/MARCQI-MICOVID19-SDOH-Manuscript.git.

## Results

Data from a total of 2,639 patients discharged between February 19 and November 19, 2020 from participating hospitals was available for this analysis. All patients had positive COVID-19 testing. Among the 2,639 patients, 42.8% were Black, 43.8% were White, 2.0% were Asian, 3.9% were Latino, and 7.5% were of unknown race and ethnicity. Significant baseline differences in the clinical characteristics of patients by race and ethnicity were observed (**Table 1**). For example, Black patients were more likely to have diabetes (42.8%), hypertension (71.9%), and moderate to severe kidney disease (32.5%) compared with White, Asian, and Latino patients (p < 0.001 for each comorbidity). Conversely, White patients were more likely to be

**Table 1. Baseline clinical characteristics and medications of patients in the MiCOVID-19 registry by race and ethnicity.**

| | Black | White | Asian | Latino | Other/ Unknown | p-value |
|---|---|---|---|---|---|---|
| No. (%) | 1129 (43) | 1157 (44) | 54 (2) | 102 (4) | 197 (7) | |
| Age. mean (SD) | 61.0 (15.9) | 68.7 (16.7) | 59.0 (15.9) | 58.6 (17.5) | 60.3 (16.3) | < 0.001 |
| Male | 567 (50) | 612 (53) | 33 (61) | 61 (60) | 109 (55) | 0.15 |
| BMI mean (SD) | 33.0 (8.9) | 30.1 (8.0) | 26.1 (5.3) | 31.8 (7.6) | 30.8 (8.5) | < 0.001 |
| Former smoker | 296 (26) | 408 (35) | 11 (20) | 33 (32) | 43 (22) | < 0.001 |
| Current smoker | 94 (8) | 82 (7) | 1 (2) | 8 (8) | 10 (5) | 0.23 |
| Diabetes | 483 (43) | 377 (33) | 20 (37) | 41 (40) | 72 (37) | < 0.001 |
| Hypertension | 812 (72) | 749 (65) | 25 (46) | 56 (55) | 102 (52) | < 0.001 |
| COPD | 121 (11) | 190 (16) | 0 (0) | 5 (5) | 8 (4) | < 0.001 |
| Asthma | 148 (13) | 130 (11) | 5 (9) | 12 (12) | 23 (12) | 0.67 |
| Other chronic lung disease | 47 (4) | 51 (4) | 1 (2) | 4 (4) | 5 (3) | 0.69 |
| Moderate to severe kidney disease | 367 (33) | 265 (23) | 7 (13) | 22 (22) | 33 (17) | < 0.001 |
| Moderate or severe liver disease | 7 (1) | 9 (1) | 1 (2) | 0 (0) | 1 (1) | 0.72 |
| Cardiovascular disease/history of myocardial infarction | 272 (24) | 366 (32) | 7 (13) | 27 (26) | 55 (28) | < 0.001 |
| Congestive heart failure/ cardiomyopathy | 160 (14) | 198 (17) | 2 (4) | 15 (15) | 18 (9) | < 0.01 |
| Cerebrovascular disease | 141 (13) | 134 (12) | 7 (13) | 11 (11) | 21 (11) | 0.92 |
| History of cancer | 80 (7) | 129 (11) | 4 (7) | 2 (2) | 3 (2) | < 0.001 |
| Dementia | 87 (8) | 211 (18) | 6 (11) | 6 (6) | 20 (10) | < 0.001 |
| HIV/AIDS | 10 (1) | 7 (1) | 0 (0) | 0 (0) | 1 (1) | 0.74 |
| Rheumatoid arthritis | 39 (3) | 38 (3) | 0 (0) | 2 (2) | 1 (1) | 0.12 |
| Organ transplant | 22 (2) | 7 (1) | 0 (0) | 2 (2) | 0 (0) | 0.01 |
| In congregate living | 154 (14) | 328 (28) | 4 (7) | 8 (8) | 29 (15) | < 0.001 |
| Known contact with an individual with COVID-19 | 270 (24) | 364 (32) | 19 (35) | 32 (31) | 52 (26) | < 0.01 |
| Healthcare worker | 69 (6) | 64 (6) | 6 (11) | 3 (3) | 10 (5) | 0.30 |
| Commercial insurance | 256 (23) | 202 (17) | 19 (35) | 18 (18) | 39 (20) | < 0.01 |
| Medicaid | 181 (16) | 93 (8) | 6 (11) | 19 (19) | 44 (22) | < 0.001 |
| Medicare | 538 (48) | 707 (61) | 14 (26) | 37 (36) | 79 (40) | < 0.001 |
| **Treatments prior to admission** | | | | | | |
| Angiotensin-converting enzyme inhibitor | 214 (19) | 230 (20) | 9 (17) | 25 (25) | 34 (17) | 0.58 |
| Corticosteroids | 145 (13) | 142 (12) | 6 (11) | 8 (8) | 21 (11) | 0.60 |
| Anticoagulants | 172 (15) | 261 (23) | 6 (11) | 15 (15) | 26 (13) | < 0.001 |
| Home oxygen | 34 (3) | 94 (8) | 0 (0) | 2 (2) | 5 (3) | < 0.001 |

Abbreviations: AIDS = acquired immunodeficiency syndrome, BMI = body mass index, COPD = chronic obstructive pulmonary disease, HIV = human immunodeficiency virus, SD = standard deviation

older (mean age 68.7 ± 16.7), have a history of smoking (35.3%), chronic obstructive pulmonary disease (COPD) (16.4%), cardiovascular disease (31.6%), congestive heart failure (17.1%), and dementia (18.2%) when compared with Black, Asian, and Latino patients (p < 0.01 for each variable). Additionally, White patients were more likely to live in a congregate setting (28.3%), such as nursing homes and assisted living facilities, compared with Black, Asian, and Latino patients (p< 0.001). Asian patients were the most likely to have reported contact with an individual with COVID-19 (35.2%), compared with White, Black, and Latino patients (p < 0.01). Black patients had the highest body mass index (BMI) on average (33.0 ± 8.9 kg/m²), compared with White (BMI 30.1 ± 8.0 kg/m²), Latino (BMI 31.8 ± 7.6 kg/m²), and Asian patients (BMI 26.1 kg/m²) (p<0.001).

## Baseline medications and laboratory values

A substantial number of patients across races and ethnicities were taking corticosteroids prior to admission (range: 7.8% in Latino patients to 12.8% in Black patients) (**Table 1**). White patients were the most likely to be receiving anticoagulation (22.6%) or home oxygen (8.1%) when compared with Black, Asian and Latino patients (p < 0.001 for each variable), potentially a reflection of a higher prevalence of underlying cardiovascular disease and COPD in this population (**Table 1**). There was no difference in angiotensin-converting enzyme inhibitor medication use between groups at baseline. This is of relevance given reports of a possible link between use of angiotensin-converting enzyme inhibitor medications and risk of infection or illness severity, though this effect has been shown to be inconsistent [33–35].

With regard to laboratory markers on admission, Asian patients had the highest baseline hemoglobin value (14.6 ± 4.3 g/dL) compared to Black, White and Latino patients (p < 0.001) (**Table 2**). Black patients had the highest baseline creatinine (2.1 ± 2.9 mg/dL) compared with White, Asian, and Latino patients (p < 0.001). Latino patients had the highest baseline bilirubin (2.8 ± 9.9 mg/dL) (p < 0.001) and lactate (3.8 ± 14.0 mmol/L) (p = 0.01) when compared with Black, White, and Asian patients. We found no differences between groups in white blood cell count or absolute lymphocyte count (**Table 2**). Of the other biomarkers for disease severity examined, creatine phosphokinase (CPK), lactate dehydrogenase (LDH), erythrocyte sedimentation rate (ESR), ferritin, and C-reactive protein (CRP) differed by race or ethnicity, with higher CPK, LDH, ferritin, CRP, and ESR values noted in Black patients (CPK, 266.0 (IQR 494.0) IU/L, LDH, 462.6 ± 318.4 IU/L, ferritin, 713.6 (IQR 1131.1) ng/mL, CRP, 51.5 ± 79.2 mg/dL, and ESR, 73.0 (IQR 39.0) mm/hour), compared with White, Latino, and Asian patients (**Table 2**).

## Patterns of clinical care

We observed that Latino patients were more often treated at smaller (< 250 bed) hospitals (61.8%) when compared with Black, White, and Asian patients (p < 0.001), whereas Black patients were more likely (7.4%) to be admitted as a transfer from another hospital when compared with White, Asian, and Latino patients (p < 0.001) (**Table 3**).

In addition to differences in baseline comorbidities, differences in inpatient treatment across groups were also observed. Specifically, Latino patients were more likely (17.6%) than other groups to receive an angiotensin converting-enzyme inhibitors (ACE-I) as an inpatient (p = 0.01) (**Table 3**). Black patients were the least likely to receive the antiviral remdesivir (2.0%, p < 0.001) and the corticosteroid dexamethasone (4.0%, p < 0.001) compared to other racial and ethnic groups (**Table 3 and Fig 1**). Although dexamethasone and remdesivir use increased over the study period, remdesivir use in Black patients remained lower at the end of the study period compared with Latino and White patients (**Fig 1**). The finding of lower remdesivir and dexamethasone use in Black patients compared with White and Latino patients was consistent amongst hospitals, including smaller (< 250 beds) versus larger (≥ 250 beds) hospitals, and teaching versus non-teaching hospitals (**S1 Table**). Asian patients most often received hydroxychloroquine (57.4%) or an IL-6 receptor inhibitor (14.8%). Latino patients most often received remdesivir (20.6%) and dexamethasone (20.6%). No difference in utilization of antibiotics was observed by race or ethnicity (**Table 3**).

White patients were more likely to have advance directives (including no resuscitation (14.3%) or mechanical ventilation (10.9%)) in place at the time of admission or to add these treatment limitations (no resuscitation (16.4%) and no mechanical ventilation (12.4%)) during the admission, compared with other groups (**Table 3**).

**Table 2. In-hospital lab values, Day 1, of patients in the MiCOVID-19 registry by race and ethnicity.**

| | Black | White | Asian | Latino | Other/ Unknown | p-value |
|---|---|---|---|---|---|---|
| No. (%) | 1129 (43) | 1157 (44) | 54 (2) | 102 (4) | 197 (7) | |
| Highest white blood cell count, K/ul, mean (SD) | 7.8 (6.4) n = 1041 | 8.1 (5.5) n = 1088 | 6.7 (3.1) n = 53 | 8.6 (4.3) n = 94 | 7.9 (4.1) n = 187 | 0.25 |
| Lowest absolute lymphocyte count, K/uL, mean (SD) | 1.8 (8.9) n = 801 | 1.6 (7.8) n = 926 | 1.5 (2.8) n = 45 | 3.5 (17.1) n = 82 | 1.2 (0.8) n = 160 | 0.37 |
| Lowest platelet count, K/uL, mean (SD) | 221.8 (88.3) n = 1038 | 215.8 (104.1) n = 1086 | 184.6 (72.0) n = 53 | 233.4 (108.2) n = 93 | 219.1 (83.4) n = 187 | 0.03 |
| Highest hemoglobin, g/dL, mean (SD) | 12.9 (2.6) n = 1040 | 13.1 (2.3) n = 1090 | 14.6 (4.3) n = 53 | 13.6 (1.9) n = 94 | 13.3 (1.8) n = 187 | < 0.001 |
| Highest creatinine, mg/dL, mean (SD) | 2.1 (2.9) n = 1043 | 1.4 (1.4) n = 1082 | 1.2 (1.0) n = 53 | 1.3 (1.4) n = 94 | 1.3 (1.2) n = 184 | < 0.001 |
| Highest ALT, IU/L, mean (SD) | 40.3 (63.5) n = 820 | 34.3 (43.0) n = 952 | 49.1 (37.2) n = 52 | 43.0 (48.8) n = 83 | 45.7 (43.6) n = 163 | 0.01 |
| Highest total bilirubin, mg/dL, mean (SD) | 0.8 (1.7) n = 803 | 0.9 (2.0) n = 942 | 0.8 (1.5) n = 52 | 2.8 (9.9) n = 82 | 0.7 (1.3) n = 160 | < 0.001 |
| Highest troponin, ng/mL, mean (SD) | 34.3 (167.1) n = 682 | 38.9 (424.9) n = 755 | 6.9 (12.6) n = 39 | 70.1 (350.2) n = 71 | 18.3 (72.6) n = 140 | 0.79 |
| Highest BNP, pg/mL, median (IQR) | 46.0 (152.0) n = 351 | 93.5 (219.0) n = 426 | 32.5 (59.5) n = 12 | 135.0 (458.0) n = 32 | 58.0 (87.0) n = 49 | < 0.001 |
| Highest CPK, IU/L, median (IQR) | 266.0 (494.0) n = 215 | 117.0 (244.5) n = 192 | 141.5 (271.0) n = 10 | 100.5 (176.5) n = 8 | 106.0 (205.0) n = 45 | < 0.001 |
| Highest LDH, IU/L, mean (SD) | 462.6 (318.4) n = 423 | 353.4 (220.9) n = 509 | 370.3 (165.7) n = 35 | 393.8 (218.9) n = 37 | 387.9 (234.3) n = 102 | < 0.001 |
| Lowest pH, median (IQR) | 7.4 (0.1) n = 348 | 7.4 (0.1) n = 327 | 7.4 (0.1) n = 28 | 7.4 (0.1) n = 24 | 7.4 (0.1) n = 53 | 0.54 |
| Highest lactate, mmol/L, mean (SD) | 2.0 (3.4) n = 591 | 1.9 (1.9) n = 719 | 1.6 (0.8) n = 45 | 3.8 (14.0) n = 54 | 2.0 (2.3) n = 125 | 0.01 |
| Highest erythrocyte sedimentation rate, mm/hour, median (IQR) | 73.0 (39.0) n = 83 | 46.5 (37.5) n = 108 | 33.0 (45.0) n = 7 | 68.0 (54.0) n = 27 | 48.0 (50.0) n = 25 | < 0.001 |
| Highest ferritin, ng/mL, median (IQR) | 713.6 (1131.1) n = 437 | 433.0 (672.0) n = 527 | 503.0 (868.8) n = 31 | 439.2 (1109.7) n = 28 | 520.6 (876.4) n = 101 | < 0.001 |
| Highest CRP, mg/dL, mean (SD) | 51.5 (79.2) n = 471 | 32.1 (55.2) n = 614 | 29.5 (48.9) n = 35 | 42.4 (65.1) n = 51 | 42.8 (75.3) n = 115 | < 0.001 |
| Highest procalcitonin, ng/mL, mean (SD) | 9.4 (76.2) n = 416 | 1.9 (20.5) n = 521 | 0.2 (0.3) n = 32 | 0.3 (0.6) n = 59 | 4.4 (31.0) n = 82 | 0.19 |
| Highest fibrinogen, mg/dL, median (IQR) | 596.0 (199.0) n = 67 | 523.0 (245.0) n = 42 | 504.5 (400.5) n = 4 | 584.0 (60.0) n = 2 | 522.5 (232.0) n = 10 | 0.30 |
| Highest Interleukin-6, pg/mL, median (IQR) | 14.1 (25.0) n = 32 | 25.8 (45.7) n = 39 | 5.0 (0.0) n = 2 | 104.7 (113.1) n = 3 | 32.9 (47.4) n = 14 | 0.06 |

Abbreviations: ALT = alanine transaminase, BNP = B-type natiuretic peptide, CPK = creatine phosphokinase, CRP = C-reactive protein, IQR = interquartile range, LDH = lactate dehydrogenase, SD = standard deviation

Asian patients most often required intensive care unit admission (33.1%) and mechanical ventilation (24.1%) compared with Black, White, and Latino patients (p = 0.04 and p < 0.01, respectively) (**Fig 2**). For those who received mechanical ventilation, Black and White patients had similar rates of being discharged alive (33.2% and 32.2%, respectively), which was lower than that seen in Asian (69.2%) and Latino (94.1%) patients (p < 0.001). At discharge, White patients were less likely to be discharged home (52.4%) and most likely to be discharged to a facility, including for hospice care (20.1%), when compared with Black, Asian, and Latino patients (p < 0.001 for each) (**Fig 2**).

## Mortality outcomes

A total of 2,160 of 2,639 patients were discharged alive (81.8%), whereas 479 (18.2%) were deceased at the time of discharge. When examining mortality by race, in-hospital mortality

**Table 3. Hospital and treatment characteristics of patients in the MiCOVID-19 registry by race and ethnicity.**

| | Black | White | Asian | Latino | Other/ Unknown | p-value |
|---|---|---|---|---|---|---|
| No. (%) | 1129 (43) | 1157 (44) | 54 (2) | 102 (4) | 197 (7) | |
| **Hospital characteristics** | | | | | | |
| Hospital bed size < 250 beds | 195 (17) | 370 (32) | 17 (31) | 63 (62) | 57 (29) | < 0.001 |
| Hospital bed size ≥ 250 beds | 934 (83) | 787 (68) | 37 (69) | 39 (38) | 140 (71) | < 0.001 |
| **Hospitalization details** | | | | | | |
| Length of stay (days), mean (SD) | 6.8 (6.5) | 7.4 (7.0) | 8.2 (5.5) | 8.1 (7.1) | 7.4 (6.5) | 0.10 |
| Transferred from another hospital | 83 (7) | 42 (4) | 2 (4) | 1 (1) | 14 (7) | < 0.001 |
| **In-hospital treatments** | | | | | | |
| Angiotensin-converting enzyme inhibitor | 112 (10) | 166 (14) | 5 (9) | 18 (18) | 29 (15) | 0.01 |
| Antibiotics | 721 (64) | 730 (63) | 29 (54) | 62 (61) | 113 (57) | 0.27 |
| Remdesivir | 25 (2) | 136 (12) | 4 (7) | 21 (21) | 19 (10) | < 0.001 |
| Hydroxychloroquine | 525 (47) | 355 (31) | 31 (57) | 22 (22) | 93 (47) | < 0.001 |
| IL-6 receptor inhibitor | 67 (6) | 60 (5) | 8 (15) | 12 (12) | 10 (5) | < 0.01 |
| Dexamethasone | 45 (4) | 166 (14) | 3 (6) | 21 (21) | 22 (11) | < 0.001 |
| Vasopressors | 207 (18) | 162 (14) | 11 (20) | 13 (13) | 32 (16) | 0.05 |
| **Treatment limitations at admission** | | | | | | |
| No cardio- pulmonary resuscitation | 46 (4) | 165 (14) | 1 (2) | 9 (9) | 7 (4) | < 0.001 |
| No mechanical ventilation | 32 (3) | 126 (11) | 1 (2) | 8 (8) | 7 (4) | < 0.001 |
| **Treatment limitations added** | | | | | | |
| Comfort care only | 66 (6) | 174 (15) | 5 (9) | 5 (5) | 11 (6) | < 0.001 |
| No cardio- pulmonary resuscitation | 138 (12) | 190 (16) | 5 (9) | 7 (7) | 22 (11) | < 0.01 |
| No mechanical ventilation | 73 (6) | 143 (12) | 4 (7) | 6 (6) | 15 (8) | < 0.001 |

Abbreviations: IL-6 = Interleukin-6, SD = standard deviation

was highest for White patients (19.6%), followed by Black patients (18.7%), Asian patients (13.0%) and Latino patients (5.9%) (p < 0.01). The mean age of death was lowest for Black patients (69.4 ± 13.3 years), a difference of greater than 8 years from White (77.9 ± 12.6 years) and Asian patients (77.6 ± 6.6 years) (p < 0.001) (**Fig 2**).

## Discussion

This study highlights key differences in baseline comorbidities, inpatient treatments, and outcomes by race and ethnicity in Michigan patients hospitalized with COVID-19. We found differences in inpatient medical therapies and use of mechanical ventilation by race and ethnicity. White patients were more likely to have advance directives in place at the time of admission or have treatment limitations added during the hospitalization. However, despite a similar age on admission to Asian and Latino patients, Black patients had high rates of inpatient mortality in the setting of COVID-19. Black patients suffered from death at a strikingly lower age than White patients, and although Black patients had higher inflammatory markers indicative of more severe illness, they were less likely to receive dexamethasone and remdesivir during the admission. While the design of this study cannot determine reasons for these differences, the findings have important ethical, moral, and policy implications.

While we confirm disparate COVID-19 outcomes based on race and ethnicity, our study adds to the knowledge base by defining that these may not only be due to variation in underlying medical comorbidities and exposure risk, but also due to differences in hospital care processes. Similar to prior studies, [14, 16, 22, 36] we found significantly higher rates of

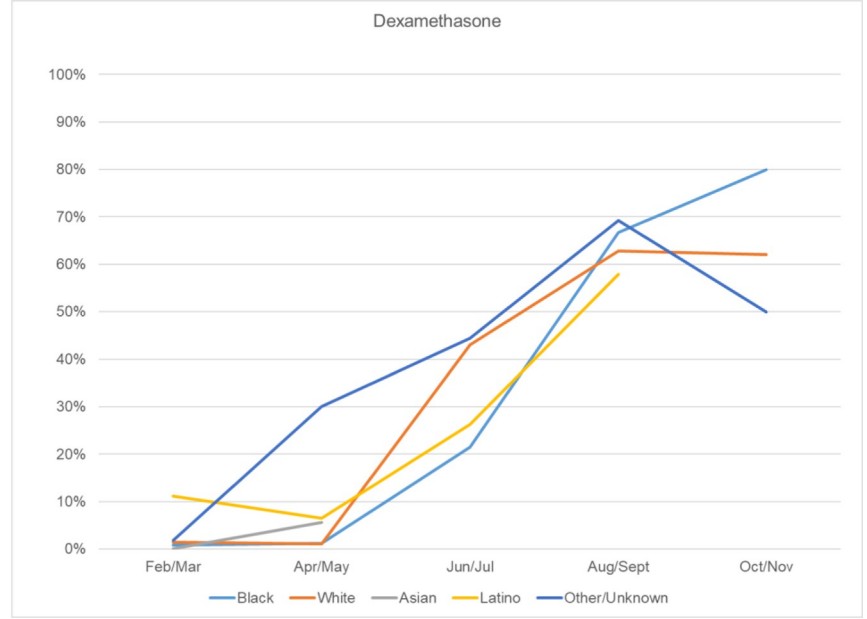

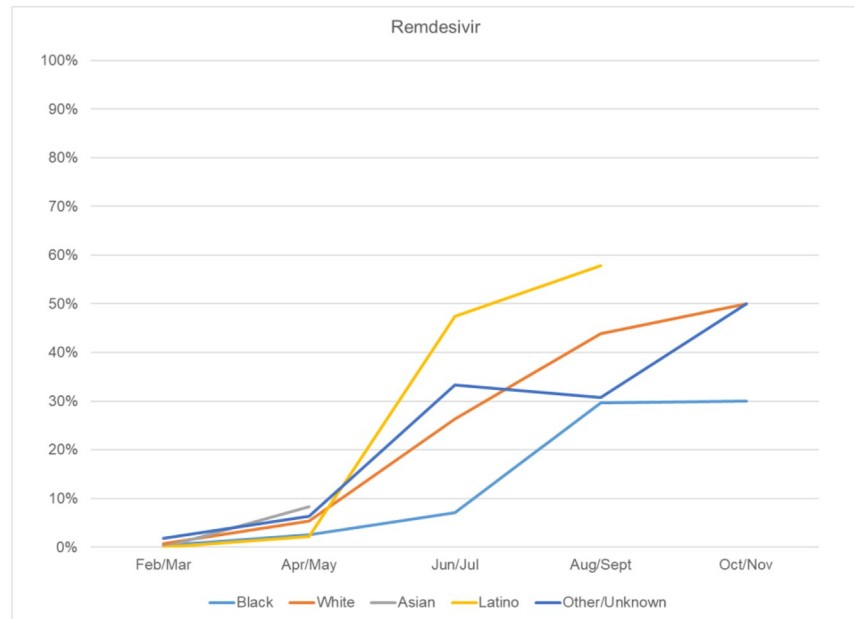

**Fig 1. In-hospital administration of dexamethasone and remdesivir by race and ethnicity.** Percent in-hospital administration of (A) the corticosteroid dexamethasone and (B) the antiviral remdesivir by race and ethnicity over the study period (February 2020 to November 2020), in 2 month increments.

hypertension, type 2 diabetes mellitus, and chronic kidney disease in Black patients, highlighting the role that these medical comorbidities play in increasing susceptibility to and mortality from COVID-19. White patients were more likely to be older, live in congregate living, had underlying cardiopulmonary disease, dementia, and have treatment limitations in place at the time of admission. Black patients were more likely to be transferred to another hospital than

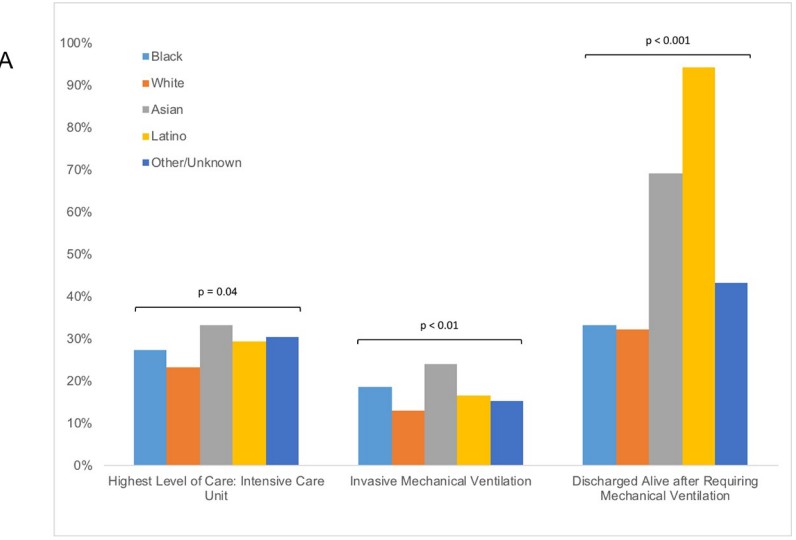

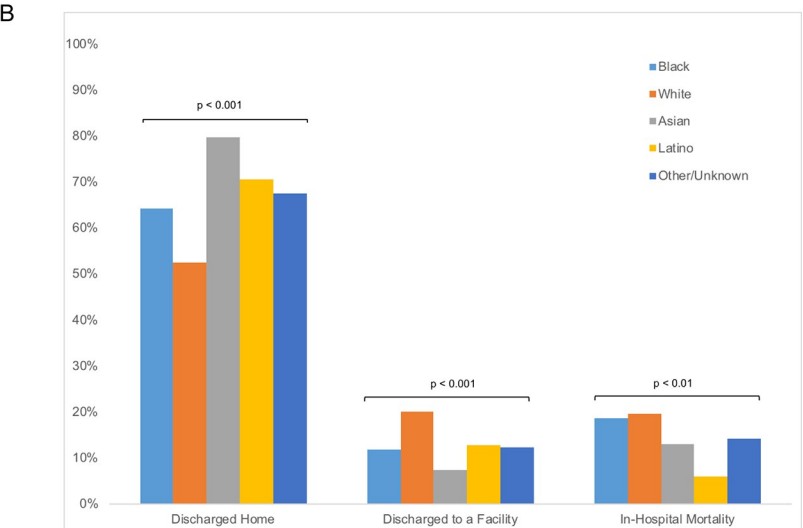

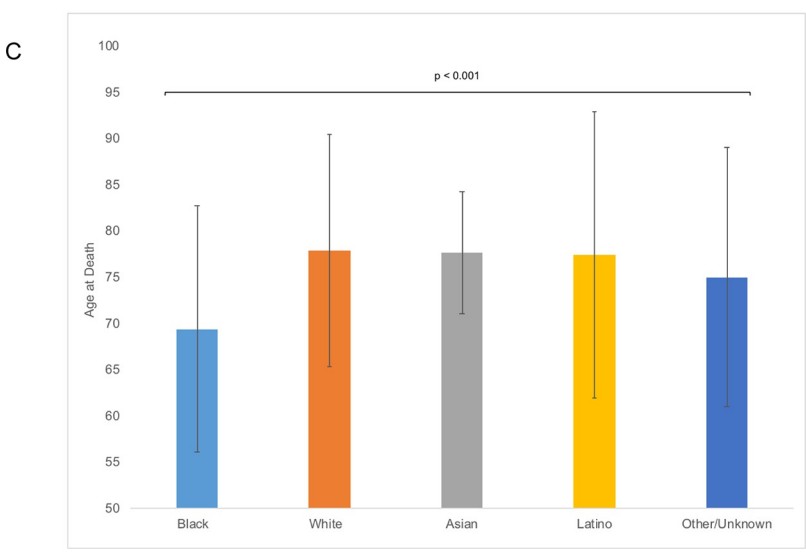

**Fig 2. In-hospital outcomes based on race and ethnicity in the MiCOVID-19 registry.** In-hospital outcomes including (A) level of hospital care required, (B) discharge location and in-hospital mortality, and (C) age at the time of death by race and ethnicity in the MiCOVID-19 Registry.

other race and ethnicity groups, likely a reflection of locally overwhelmed health systems potentially explaining delays or absence of COVID-19 specific treatment. Conversely, Latino patients were more likely to be admitted to a smaller (<250 bed) hospital compared with other racial and ethnic groups, receive corticosteroids and remdesivir, be discharged alive after requiring mechanical ventilation, and had a lower associated mortality in this study. Importantly, despite these differences, non-Latino White and Black patients had the highest in-hospital mortality rate at 19.6% and 18.7%, respectively. These data suggest that age, pre-existing disparities in health among racial and ethnic groups, and care processes directly influenced COVID-19 outcomes.

Besides comorbidities, it is plausible that differential community, facility, or occupational exposure to SARS-CoV-2; population density; access to preventive medical care and management of underlying risk factors; and other social factors play critical roles in preventing severe COVID-19 and death. This theory is supported by a recent single-center study of patients in Michigan noting that residing in a densely populated area was associated with an increased risk of hospitalization for COVID-19 [37]. Similar to the findings reported here, this recent study suggested no significant differences in a need for ICU admission by race. Our findings are consistent with two single-center studies that did not find a significant increase in mortality in Black patients hospitalized with COVID-19 compared with other racial groups [37, 38]. However, our study adds significantly by pointing out that while the mortality rate may not differ, Black patients died at a strikingly lower age than any other racial or ethnic group and were less likely to receive medications known to have efficacy for COVID-19 [39, 40]. By the time an individual is hospitalized with COVID-19, the many facets of socioeconomic disparity and systemic racism may have already heavily impacted the individual's risk for poor outcomes, which may not be overcome by even superb inpatient critical care.

We observed that White patients were more likely to have advance directives in place at the time of admission. In part, this may reflect demographic differences; for instance, White patients were older at admission and more likely to be admitted from congregate living (such as a long-term skilled nursing facility) than others. However, disparities in advance directives for Black patients have also been attributed to potential distrust in the healthcare system and poor access to knowledge on how complete the required documents [41–43]. Information on advance directives should be made available to patients in the course of their routine health maintenance, and healthcare systems should be equipped to provide additional resources on information on planning at the time of admission and upon request.

Our findings should be interpreted in the context of the following limitations. This is an observational analysis, and therefore, we cannot determine causality. Because the registry only collects data on patients who are hospitalized, there is a possibility that true outcomes differences could be over- or under-estimated if patients did not come to the hospital for evaluation or were not admitted. Data on race and ethnicity were drawn from the medical record; thus, the validity of these data are unknown. Finally, we are unable to discern whether variations in clinical care observed relate to differential treatment of individuals by race, or whether these reflect hospital level factors such as access to treatments or protocols that were changing over time. Since more Black patients were transferred for care, it is possible that care variation represents hospital rather than patient level factors.

## Conclusions

These data from the MiCOVID-19 Registry, inclusive of 38 non-federal hospitals in the State of Michigan, demonstrate similar COVID-19 mortality in Black and White patients, though Black patients died at a much younger age than other race and ethnic groups. We observed differences in hospital care, suggesting that health status and susceptibility to COVID-19 mortality could be further compounded by inpatient treatment factors.

## Supporting information

**S1 Table. In-hospital treatment for COVID-19 with remdesivir and dexamethasone by race and ethnicity, stratified by hospital size, teaching versus non-teaching hospital, and the surrounding population density of the hospital.**
(DOCX)

## Author Contributions

**Conceptualization:** Nadia R. Sutton, Sheria G. Robinson-Lane.

**Formal analysis:** Heather A. Chubb.

**Funding acquisition:** Vineet Chopra.

**Investigation:** Sheria G. Robinson-Lane, Vineet Chopra.

**Methodology:** Nadia R. Sutton, Heather A. Chubb.

**Project administration:** Tae Kim.

**Supervision:** Nadia R. Sutton, Vineet Chopra.

**Writing – original draft:** Nadia R. Sutton, Raymond Y. Yeow, Vineet Chopra.

**Writing – review & editing:** Nadia R. Sutton, Sheria G. Robinson-Lane, Raymond Y. Yeow, Heather A. Chubb, Vineet Chopra.

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
