## [Decision Letter · Decision Letter 0]

1 Dec 2021

PONE-D-21-24697Racial and Ethnic Variation in COVID-19 Care, Treatment, and Outcomes: A Prospective Cohort Study from the MiCOVID-19 RegistryPLOS ONE

Dear Dr. Sutton,

Thank you for submitting your manuscript to PLOS ONE. After careful consideration, we feel that it has merit but does not fully meet PLOS ONE’s publication criteria as it currently stands. Therefore, we invite you to submit a revised version of the manuscript that addresses the points raised during the review process.

We look forward to receiving your revised manuscript.

Kind regards,

Giordano Madeddu

Academic Editor

PLOS ONE

Journal Requirements:

2. Thank you for stating the following in the Competing Interests/Financial Disclosure* (delete as necessary) section:

“N.R.S. has received honoraria for speaking from Zoll Medical and is a Cordis and Philips advisory board member. V.C. has received royalties from Wolters Kluwer publishing, Oxford University publishing, and Lippincott for sales of academic textbooks. All other authors have no disclosures.”

We note that one or more of the authors are employed by a commercial company: Zoll Medical a. Please provide an amended Funding Statement declaring this commercial affiliation, as well as a statement regarding the Role of Funders in your study. If the funding organization did not play a role in the study design, data collection and analysis, decision to publish, or preparation of the manuscript and only provided financial support in the form of authors' salaries and/or research materials, please review your statements relating to the author contributions, and ensure you have specifically and accurately indicated the role(s) that these authors had in your study. You can update author roles in the Author Contributions section of the online submission form.

Reviewers' comments:

Reviewer's Responses to Questions

**Comments to the Author**

1. Is the manuscript technically sound, and do the data support the conclusions?

Reviewer #1: Partly

Reviewer #2: Yes

2. Has the statistical analysis been performed appropriately and rigorously? 

Reviewer #1: No

Reviewer #2: Yes

3. Have the authors made all data underlying the findings in their manuscript fully available?

Reviewer #1: No

Reviewer #2: Yes

4. Is the manuscript presented in an intelligible fashion and written in standard English?

Reviewer #1: Yes

Reviewer #2: Yes

5. Review Comments to the Author

Reviewer #1: It is unclear why the authors referred to racial/race and ethnic/ethnicity, not only ethnic/ethnicity. Therefore, I suggest removing racial and race from the manuscript.

Introduction

Add “Introduction” at page 5.

Authors wrote “severe acute respiratory syndrome coronavirus 2 (SARS-CoV-2), also known as COVID-19” SARS-COV-2 is the virus, COVID-19 is the disease, so writing “also known as COVID-19” is incorrect. Please reformulate the sentence, also adding the meaning of COVID-19. In the all manuscript the authors confused COVID-19 with SARS-CoV-2 (e.g. at the page 7 the authors wrote “Patients were included in the registry if they tested positive for COVID-19”. In this case, they should use SARS-CoV-2). Please re-read the manuscript carefully and correct all of the mistakes.

Please update the number of deaths (4’760’000 the 26th of September).

Before writing about the racial and ethnic disparities, I suggest writing about what COVID-19 is, and its presentation. So, I suggest explaining the symptoms, both major (fever, dyspnea, cough) and minor (dysgeusia, anosmia, gastrointestinal symptoms, headache and skin lesions), giving the reader the full picture of the diseases. Then, I suggest reading these articles that you could use and cite in the manuscript expanding this part (https://doi.org/10.1371/journal.pone.0248009;
https://doi.org/10.1097/IPC.0000000000000952, https://doi.org/10.1002/hed.26269).

The sentence “Despite these theories, few studies have examined differences in hospital care processes as a potential moderator of adverse outcomes in patients with COVID-19”. It is not clear what the authors mean with “moderator”. Please comment.

Methods

It is not clear if the authors conducted a retrospective or a prospective study. In the title, they wrote “prospective” and so in the methods section; contrary to the abstract, they wrote, “ This was a retrospective cohort study using MiCOVID-19, a multi-center, prospective, collaborative quality improvement registry”.”

In my opinion, they have conducted a retrospective study, so I suggest removing “prospective” from the title and methods.

The authors wrote “Patients discharged from a participating site between February 19, 2020 and November 19, 2020 were used for this analysis.” What is it the definition of “discharged”. Did the authors also consider dead patients?

What is it a “trained clinical abstractor”?

The authors wrote “when there was a strong documented clinical suspicion of COVID-19 but testing was not performed or recorded due to logistical constraints”. Please specify what “document clinical suspicion” means. Furthermore, I suggest writing how many patients responded to this definition and how patients had a positive swab.

Regarding ethnicity, it is unclear what they meant with “Non-Hispanic”. Please comment.

Results

In the results section, the authors wrote “42.7% were Black, 43.8% were White, 2.0% were Asian, 3.9% were Latino, and 7.5% were of unknown race and ethnicity.” Thus, the sum is 99.9%.

Have you evaluated the distribution of the different values? For example, creatinine, bilirubin and lactate seem having a non normal distribution. If the value does not have a normal distribution, you should use the median and interquartile range, not mean ± SD. Furthermore, ANOVA could not be used. Please verify the distribution of all different variables, and re-perform the analysis.

I suggest performing a multivariate analysis to evaluate if ethnic groups, comorbidity, or treatment are associated with an increased risk of death.

Discussion

Please modify the discussion with the new results.

Regarding chronic treatment, assuming angiotensin receptor blockers was associated with an increased risk of infection (10.26355/eurrev_202101_24424) and a worse outcome (10.1186/s13643-021-01802-6). I suggest adding a sentence about it and these two references.

Tables

-Legend with the meaning of the abbreviations is missing. Please add it.

The tables' title are not clear. Please re-write the title explain to the reader what information are present in each table.

Furthermore, I suggest adding the “overall” column.

Reviewer #2: Manuscript is technically sound for analysis.

Statistical analysis was performed nicely all analysis ,including table, graph , figure are nicely explained.

All data are available. Its written in standard English.

6. PLOS authors have the option to publish the peer review history of their article (what does this mean?). If published, this will include your full peer review and any attached files.

Reviewer #1: No

Reviewer #2: **Yes: **S M Habibur Rahman Habib

---

## [Author Response · Author response to Decision Letter 0]

12 Jul 2022

Response to editor comments

We would like to respond to the question from the Editorial Board regarding employment of an author by a commercial company: Zoll Medical. We would like to clarify that Dr. Nadia Sutton is not employed by Zoll Medical. Rather, Dr. Sutton is employed fully by the University of Michigan. Zoll Medical had no role in the study design, data collection or analysis, decision to publish, preparation of the manuscript and does not provide any salary support or effort toward any investigator involved in the study, and also did not provide any research materials. The relationship is not relevant to the content of the manuscript and was included in the Competing Interests section only for completeness and full transparency. Our updated Funding Statement and Competing Interests Statements are as follows: 

Funding Statement: Blue Cross Blue Shield of Michigan (BCBSM) and Blue Care Network as part of the BCBSM Value Partnerships program. The funder, the Blue Cross Blue Shield of Michigan and Blue Care Network, provided support in the form of salaries for authors (H.A.C. and T.K.), but did not have any additional role in the study design, data collection and analysis, decision to publish, or preparation of the manuscript. The specific roles of these authors are articulated in the ‘author contributions’ section. The National Institute on Aging 1K76AG064426-01A1 (N.R.S.) and K01AG06542001A1 (S.G.R.).

Competing Interests Statement: N.R.S. has received honoraria for speaking from Zoll Medical and is a Cordis and Philips advisory board member. This does not alter our adherence to PLOS ONE policies on sharing data and materials. V.C. has received royalties from Wolters Kluwer publishing, Oxford University publishing, and Lippincott for sales of academic textbooks. All other authors have no disclosures. 

Response to reviewer comments

Reviewer #1

It is unclear why the authors referred to racial/race and ethnic/ethnicity, not only ethnic/ethnicity. Therefore, I suggest removing racial and race from the manuscript.

Thank you for this suggestion and for allowing us to clarify this concern. The MiCOVID-19 Registry reflects data that is captured by the electronic health records in the State of Michigan, in the United States of America. In the USA, health care demographic form fields usually follow the same categorization of race and ethnicity as is used by the United States Census Bureau. The US Census Bureau follows standards on race and ethnicity set by the U.S. Office of Management and Budget in 1997. These guide how the federal government collects and presents data on these topics. Data is collected on race and ethnicity (Hispanic or Latino origin) in two separate questions. 5 categories of race are utilized: White, Black or African American, American Indian or Alaska Native, Asian, or Native Hawaiian or Other Pacific Islander. We have therefore collected data in accordance with standard practice. 

Add “Introduction” at page 5.

We have added “Introduction” on page 5.

Authors wrote “severe acute respiratory syndrome coronavirus 2 (SARS-CoV-2), also known as COVID-19” SARS-COV-2 is the virus, COVID-19 is the disease, so writing “also known as COVID-19” is incorrect. Please reformulate the sentence, also adding the meaning of COVID-19. In the all manuscript the authors confused COVID-19 with SARS-CoV-2 (e.g. at the page 7 the authors wrote “Patients were included in the registry if they tested positive for COVID-19”. In this case, they should use SARS-CoV-2). Please re-read the manuscript carefully and correct all of the mistakes.

We have modified the text in the Introduction, Methods, and Discussion to clarify that SARS-CoV-2 is the virus that causes the disease COVID-19. 

Please update the number of deaths (4’760’000 the 26th of September).

We have updated the number of deaths to reflect the most recent statistics available.

Before writing about the racial and ethnic disparities, I suggest writing about what COVID-19 is, and its presentation. So, I suggest explaining the symptoms, both major (fever, dyspnea, cough) and minor (dysgeusia, anosmia, gastrointestinal symptoms, headache and skin lesions), giving the reader the full picture of the diseases. Then, I suggest reading these articles that you could use and cite in the manuscript expanding this part (https://doi.org/10.1371/journal.pone.0248009;
https://doi.org/10.1097/IPC.0000000000000952, https://doi.org/10.1002/hed.26269).

We have added a statement describing clinical manifestations of COVID-19 to the introduction and have included citations for the articles that the reviewer recommended. However, we understand that this is likely common knowledge at this stage. 

The sentence “Despite these theories, few studies have examined differences in hospital care processes as a potential moderator of adverse outcomes in patients with COVID-19”. It is not clear what the authors mean with “moderator”. Please comment.

In this context, the term “moderator” is used to suggest that hospital care processes may be moderated by, impact, change, or influence the outcomes of patients with COVID-19. 

Methods

It is not clear if the authors conducted a retrospective or a prospective study. In the title, they wrote “prospective” and so in the methods section; contrary to the abstract, they wrote, “ This was a retrospective cohort study using MiCOVID-19, a multi-center, prospective, collaborative quality improvement registry”.”

In my opinion, they have conducted a retrospective study, so I suggest removing “prospective” from the title and methods.

The study was designed in March of 2020 (at the outset of the pandemic), to collect data on patients with COVID-19 who would be hospitalized in the future. Data was collected prospectively on all admissions. However, to compile this analysis, we looked at previously collected data. Therefore, we agree that it would be reasonable to use the term retrospective in most instances in the text. We have modified the manuscript accordingly. 

The authors wrote “Patients discharged from a participating site between February 19, 2020 and November 19, 2020 were used for this analysis.” What is it the definition of “discharged”. Did the authors also consider dead patients?

Discharge refers to patients leaving the facility - alive or dead. On page 12, it is noted that “A total of 2,160 of 2,639 patients were discharged alive (81.8%), whereas 479 (18.2%) were deceased at the time of discharge.”

What is it a “trained clinical abstractor”?

In the MiCOVID-19 data collection, we recruited abstractors at participating hospitals who are individuals that performed manual data abstraction from patient records for the purpose of this research study. These individuals were then trained on how to perform data abstraction from patient records using a standardized template. Many of these individuals had a clinical background, such as being a registered nurse. 

The authors wrote “when there was a strong documented clinical suspicion of COVID-19 but testing was not performed or recorded due to logistical constraints”. Please specify what “document clinical suspicion” means. Furthermore, I suggest writing how many patients responded to this definition and how patients had a positive swab. 

At the outset of the pandemic, testing for COVID-19 was scarce and thus many patients were presumed to have the disease without a confirmatory test. For our study, when there was a diagnosis of COVID-19 documented but no testing confirming infection, we categorized such cases as “strong documented clinical suspicion of COVID-19.” In other words, if the treating provider suspected the patient had COVID-19 and made a presumptive diagnosis without confirmatory testing, that was considered to be a strong documented clinical suspicion. However, for this study, only patients with a positive swab were included (n = 2369). 

Regarding ethnicity, it is unclear what they meant with “Non-Hispanic”. Please comment.

As above, these are the categories of ethnicity utilized by the US Census Bureau. According to the US Census Bureau, “Hispanic origin can be viewed as the heritage, nationality, lineage, or country of birth of the person or the person’s parents or ancestors before arriving in the United States. People who identify as Hispanic, Latino, or Spanish may be any race.” US federal agencies must use a minimum of two categories for ethnicity: "Hispanic or Latino" and "Not Hispanic or Latino."

Results

In the results section, the authors wrote “42.7% were Black, 43.8% were White, 2.0% were Asian, 3.9% were Latino, and 7.5% were of unknown race and ethnicity.” Thus, the sum is 99.9%.

We thank the reviewer for pointing this out. In this study, 42.8% of patients were Black. The value has been corrected in the text of the manuscript.

Have you evaluated the distribution of the different values? For example, creatinine, bilirubin and lactate seem having a non normal distribution. If the value does not have a normal distribution, you should use the median and interquartile range, not mean ± SD. Furthermore, ANOVA could not be used. Please verify the distribution of all different variables, and re-perform the analysis.

We appreciate the suggestion. These data are descriptive and we intend for them to be informational rather than inferential. We did discuss this with our statistical authors on the paper, who added that ANOVA is not very sensitive to moderate deviations from normality and simulation studies have shown that the false positive rate is not affected much by this assumption violation.[1-3] Essentially, when you take a large number of samples from a population, the means are approximately normally distributed even when the population is not normal. It is generally thought that a sample size of 30 can be considered adequate in generalizing to a standard normal distribution. There are a few lab values where the sample sizes in the Latino and Asian groups are less than 30, and it may be more appropriate to report the median and IQR for these lab values and use the Kruskal-Wallis non-parametric test to compare them. The methods have been updated to reflect this, and Table 2 has been updated to reflect the changed analysis (median and IQR with a Kruskal-Wallis p-value for all lab values where one of the groups had a sample size less than 30). With this new analysis, we now found significant differences in the highest BNP and the highest CPK, with White and Latino patients having higher BNP values compared with other groups, and Black patients having higher CPK values compared with other groups. 

I suggest performing a multivariate analysis to evaluate if ethnic groups, comorbidity, or treatment are associated with an increased risk of death.

To respond to Reviewer #1’s suggestion, we created a logistic model for the outcome of death while adjusting for patient factors, comorbidities, and treatments, similar to models used for studies from the MiCOVID-19 Registry and others.[4-6] COVID-19 specific treatments were defined as hydroxychloroquine, combination hydroxychloroquine plus azithromycin, vitamin C (oral or intravenous), interleukin (IL)-6 inhibitors, remdesivir, or tocilizumab. 

The odds ratio and 95% confidence intervals for the final model are listed below.

 Odds Ratio 95% Confidence Interval

Age 1.05 (1.04, 1.06)

Black compared with White 1.36 (1.07, 1.72)

Asian compared with White 0.98 (0.40, 2.37)

Latino compared with White 0.40 (0.17, 0.97)

Other/Unknown compared with White 1.04 (0.65, 1.66)

Never Smoked 0.73 (0.58, 0.91)

>3 Comorbidities 1.61 (1.27, 2.04)

ARB 0.48 (0.32, 0.72)

ACE-I 0.45 (0.31, 0.67)

Antibiotics 2.54 (1.93, 3.34)

COVID-specific treatment 1.37 (1.08, 1.74)

Statins 0.73 (0.58, 0.92)

Steroids 1.56 (1.24, 1.96)

It is our interpretation that these data do not alter the major findings of our manuscript, and so we have elected to not addend the manuscript with this additional data at this time. We point out though that our findings regarding mortality increasing with age, certain race categories, comorbidities and use of steroids in a pre-RECOVER trial era are consistent with other reports.[7] Hence, while we considered adding these data in, we did not think these findings were novel enough to be included. However, if the editors feels differently, we are happy to revisit this request. 

Discussion

Please modify the discussion with the new results.

The discussion has been updated where needed.

Regarding chronic treatment, assuming angiotensin receptor blockers was associated with an increased risk of infection (10.26355/eurrev_202101_24424) and a worse outcome (10.1186/s13643-021-01802-6). I suggest adding a sentence about it and these two references.

Thank you for the suggestions. We have added a statement about baseline use of angiotensin-converting enzyme inhibitor medications in our results section and have added the references suggested by the reviewer. We will add that the most recent data has not shown for this effect to be consistent. 

Tables

Legend with the meaning of the abbreviations is missing. Please add it.

We have added a list of abbreviations after each table. Thank you for the suggestion.

The tables' title are not clear. Please re-write the title explain to the reader what information are present in each table.

We had added more detail to the Table titles.

Furthermore, I suggest adding the “overall” column.

We can appreciate the reviewer’s interest in this additional data column. However, as the tables already have 7 columns due to the different race and ethnicity categories, we are concerned that the column width will become cumbersome for readers of the manuscript. If the editor wishes for an “overall” column to be added prior to publication, we are happy to provide summary statistics. 

Reviewer #2

Manuscript is technically sound for analysis.

Statistical analysis was performed nicely all analysis ,including table, graph , figure are nicely explained.

All data are available. Its written in standard English.

The authors thank Reviewer #2 for these positive remarks and feedback. 

References

1. Glass G, Peckham P, Sanders J. Consequences of failure to meet assumptions underlying fixed effects analyses of variance and covariance. Rev. Educ. Res. 1972;42: 237-288.

2. Harwell M, Rubinstein E, Hayes W, Olds C. Summarizing Monte Carlo results in methodological research: the one- and two-factor fixed effects ANOVA cases. J. Educ. Stat. 1992;17: 315-339.

3. Lix L, Keselman J, Keselman H. Consequences of assumption violations revisited: A quantitative review of alternatives to the one-way analysis of variance F test. Rev. Educ. Res. 1996;66: 579-619.

4. Tipirneni R, Karmakar M, O'Malley M, Prescott HC, Chopra V. Contribution of Individual- and Neighborhood-Level Social, Demographic, and Health Factors to COVID-19 Hospitalization Outcomes. Ann Intern Med. 2022;175(4):505-12. Epub 2022/02/22. doi: 10.7326/M21-2615. PubMed PMID: 35188790; PubMed Central PMCID: PMCPMC8982172.

5. Chopra V, Flanders SA, Vaughn V, Petty L, Gandhi T, McSparron JI, et al. Variation in COVID-19 characteristics, treatment and outcomes in Michigan: an observational study in 32 hospitals. BMJ Open. 2021;11(7):e044921. Epub 2021/07/25. doi: 10.1136/bmjopen-2020-044921. PubMed PMID: 34301650; PubMed Central PMCID: PMCPMC8313307.

6. Imam Z, Odish F, Gill I, O'Connor D, Armstrong J, Vanood A, et al. Older age and comorbidity are independent mortality predictors in a large cohort of 1305 COVID-19 patients in Michigan, United States. J Intern Med. 2020;288(4):469-76. Epub 2020/06/05. doi: 10.1111/joim.13119. PubMed PMID: 32498135; PubMed Central PMCID: PMCPMC7300881.

7. Group RC, Horby P, Lim WS, Emberson JR, Mafham M, Bell JL, et al. Dexamethasone in Hospitalized Patients with Covid-19 - Preliminary Report. N Engl J Med. 2020. Epub 2020/07/18. doi: 10.1056/NEJMoa2021436. PubMed PMID: 32678530; PubMed Central PMCID: PMCPMC7383595.

---

## [Decision Letter · Decision Letter 1]

14 Oct 2022

Racial and Ethnic Variation in COVID-19 Care, Treatment, and Outcomes: A Retrospective Cohort Study from the MiCOVID-19 Registry

PONE-D-21-24697R1

Dear Dr. Sutton,

We’re pleased to inform you that your manuscript has been judged scientifically suitable for publication and will be formally accepted for publication once it meets all outstanding technical requirements.

Kind regards,

Zivanai Cuthbert Chapanduka, MBChB (M.D)

Academic Editor

PLOS ONE

Additional Editor Comments (optional):

Reviewers' comments:

Reviewer's Responses to Questions

**Comments to the Author**

1. If the authors have adequately addressed your comments raised in a previous round of review and you feel that this manuscript is now acceptable for publication, you may indicate that here to bypass the “Comments to the Author” section, enter your conflict of interest statement in the “Confidential to Editor” section, and submit your "Accept" recommendation.

Reviewer #1: All comments have been addressed

2. Is the manuscript technically sound, and do the data support the conclusions?

Reviewer #1: Yes

3. Has the statistical analysis been performed appropriately and rigorously? 

Reviewer #1: Yes

4. Have the authors made all data underlying the findings in their manuscript fully available?

Reviewer #1: Yes

5. Is the manuscript presented in an intelligible fashion and written in standard English?

Reviewer #1: Yes

6. Review Comments to the Author

Reviewer #1: The authors replied to all my comment. In my opinion the manuscript is now suitable to be published.

7. PLOS authors have the option to publish the peer review history of their article (what does this mean?). If published, this will include your full peer review and any attached files.

Reviewer #1: No

---

## [Editor Report · Acceptance letter]

21 Oct 2022

PONE-D-21-24697R1 

Racial and Ethnic Variation in COVID-19 Care, Treatment, and Outcomes: A Retrospective Cohort Study from the MiCOVID-19 Registry 

Dear Dr. Sutton:

I'm pleased to inform you that your manuscript has been deemed suitable for publication in PLOS ONE. Congratulations! Your manuscript is now with our production department. 

Kind regards, 

on behalf of

Dr. Zivanai Cuthbert Chapanduka 

Academic Editor

PLOS ONE